# Relationship between Different Dimensions of Workplace Spirituality and Psychological Well-Being: Measuring Mediation Analysis through Conditional Process Modeling

**DOI:** 10.3390/ijerph191811244

**Published:** 2022-09-07

**Authors:** Rizwan Raheem Ahmed, Farwa Abbas Soomro, Zahid Ali Channar, Alharthi Rami Hashem E, Hassan Abbas Soomro, Munwar Hussain Pahi, Nor Zafir Md Salleh

**Affiliations:** 1Faculty of Management Sciences, Indus University, Gulshan 17, Karachi 75300, Pakistan; 2Faculty of Management Sciences, Shaheed Zulfikar Ali Bhutto Institute of Science and Technology, Karachi 75600, Pakistan; 3Department of Business Administration, Sindh Madressatul Islam University, Karachi 74000, Pakistan; 4Department of Financial and Administrative Sciences, Ranyah University College, Taif University, P.O. Box 11099, Taif 21944, Saudi Arabia; 5Department of Business Administration, Sukkur IBA, Sukkur 65200, Pakistan; 6School of Business Management, Universiti Utara Malaysia, Sintok 06010, Malaysia; 7Azman Hashim International Business School, Universiti Teknologi Malaysia, Skudai 18310, Malaysia

**Keywords:** workplace spirituality, psychological well-being, job stress, spirituality orientation, university teachers, structural equation modeling

## Abstract

The present study aims to identify the relationship between workplace spirituality, compassion, relationship with others at work, spiritual orientation, organizational value and alignment of personal values, and psychological well-being among universities’ teachers. Further, the mediating effect of job stress was also identified between workplace spirituality and psychological well-being. Similarly, the mediation of environmental mastery between organizational values and alignment of personal values and psychological well-being was examined. Finally, we examined the mediation of personal growth between spiritual orientation and psychological well-being. The data were collected through a structured and adapted five-point Likert scale using a purposive sampling technique, with a total sample of 873 male and female university teachers working in the private and government universities. We employed structural equation modeling to check the relationship among the considered variables for analysis purposes. The results show a strong positive relationship between the independent and dependent variables. The findings further demonstrated that the mediation analysis confirms that job stress mediates the relationship between workplace spirituality and psychological well-being, and environmental mastery mediates between organizational values and alignment of personal values and psychological well-being. Finally, personal growth mediates between spiritual orientation and psychological well-being. To maintain the excellent quality of education, educational institutes need to identify and imply the practice of workplace spirituality that will help to reduce job stress and improve the psychological well-being of universities’ teachers, thus resulting in better educational output.

## 1. Introduction

The experience of teaching at the university level is quite stimulating. It carries an enriched experience full of information and is also socially quite fascinating, but it also comes at a cost. At times, this role becomes a source of competition and thus becomes more stressful than any other job. Therefore, it can be concluded that the pursuit of a long and successful career even in academia is challenging and arduous, making faculty members vulnerable to risks of compromised well-being [1,2], because faculty members within their role have several other responsibilities, like undertaking research, administrative tasks, winning external grant projects, and teaching the most important. It, coupled with working extra time for other administrative or research-related jobs, with no such significant salary increases, makes it even more challenging [3]. It is a long-ignored reality that these faculty members are at risk of compromised well-being [4]. Further, the stress is added upon by the conflicts and ambiguity in the roles assigned to these teachers [5], making it challenging to maintain a work-life balance [6]. It has been evident that illness, absence, high turnover rates, early retirement, and mental issues are majorly caused by the stress these faculty members undergo [7]. Researchers undertaking the study to understand the professional health of these faculty members have laid great emphasis upon the fact that high level of stress, lower level of stress resistance, and even lowest level of tolerance in frustrating situations is much higher in the social group of teachers in comparison to that of many other social and professional groups [8]. Conventionally, the teachers are at the forefront of nurturing and developing students’ potential [9], and teachers help students’ growth [10]. Teachers must remain physically and mentally well to help students grow and reach their full potential [11]. A great deal of literature has explained the interaction of social and health conditions of academic staff members [12]. Part of the work is devoted to improving the quality of education through the improvement of the working conditions of teachers. More recently, studies on the creating and building of a sustainable and healthy working environment during the contractual, partial work setting [13] and extending to the aspect of stress and struggle an employee goes through during the job [14] seem to be the relevant ones. Faculty members teaching at such a high level also feel exhaustion [15], stress, and depression [16] prevail, along with low work–life balance [4].

The manifestation of a social welfare structure depends on a network representing a definition of health, satisfaction with physical health conditions, and social interactions. Accordingly, three basic norms define the rule and law of positive emotions, which are found in the welfare system and its structure [1]. Workers’ lives are “One of the most prominent challenges posed to the leaders these days” [17]. With the increased and more devoted focus on the employee’s well-being by the researchers, this issue is being highlighted and has gained importance in discussions [3]. Therefore, “Academic interest in employee welfare has also increased significantly in recent years” [3]. Similarly, “the well-being of the workforce has emerged as a crucial component in good governance-based research” [16]. Before understanding what well-being implies to teachers, we must define this term. Well-being refers to being tested for one’s belief about a happy and satisfying life. It has also been described as a social environment in which people understand and learn about their self-capabilities, understand and face the day-to-day pressures, their official duties, tasks and responsibilities along with being a responsible and committed member of the community” [4]. A person’s “mental well-being” is understood to be his/her positive experience, which is primarily determined by factors of external influence [18]. Psychological well-being can be understood by analyzing the factors such as performance, job satisfaction, and experience. Various factors can be linked to employee well-being, like physical, mental, and social [19]. Employee well-being is essential to organizations because it reflects the organization’s life. For the well-being of the employees, organizations must be healthy and healthy organizations are well managed and organized and must have standardized efforts to enhance the psychological well-being and overall productivity of their employees by providing them practical purpose, organizational support, and equal opportunities for professional growth and better personal life [20]. Thus, it is vital to improving the well-being of employees. However, employee welfare takes many shapes and actions according to an organization’s policies and the implementation of these policies to increase the employee’s overall well-being can improve one type of employee well-being while reducing a different kind of employee well-being. Therefore, it is crucial to see if a factor in the workplace can improve their quality of life without harming any other type of employee well-being [1]. The spirituality of the workplace is one aspect of the workplace that can enhance many types of employee well-being. The level of spirit defines a person’s expression or spiritual knowledge at work puts at their job [21]. Specifically, work ethic refers to a meaningful employee experience and community at work [22]. Other related words are associated with spiritual experience in the workplace, including calling, membership [17], and transcendence [19]. Existing literature raises the need for workplace spirituality [13], and that is why one of the significant aspects of spirituality and work has been described and termed as one of the most attention-seeking issues in terms of scientific research [23].

According to the literature, workplace spirituality “tends to have a strong impact and effects on the individuals, organizations, and community’s overall well-being and prosperity” [23]. Additionally, workplace spirituality can help these organizations that address the general health issues related to workplace spirituality that affects the employed human resource [24]. Similarly, it suggests that spirituality improves general employee health and tends to improve the employee’s overall well-being. Other literature also shows that there is still a great deal, and this should be debated that this practice at the organizational level may improve factors like increased ethics, decreased stress, and less burnout at work [25]. In line with this, a model of the concept was developed where the work ethic, through the mediation of the organization’s commitment to employees, is linked to the mental well-being of employees [26]. The spiritual foundation in the office environment depends on the employee’s belief in the purpose and meaningful work [27]. Suppose the content of the work gives people a good spiritual experience. In that case, it will lead to spiritual growth and development, empathy and feelings of happiness, motivation for work, and well-being [1]. It has been evident in the literature that workplace spirituality is related to and consequent upon the feeling of being satisfied by passing on to individuals, groups, or organizations [12]. The main aspects of the work ethic mentioned are internal health, purpose, and purpose in the workplace, a feeling of social togetherness, and alignment with the organization’s values [17]. Feeling part of the community (connecting with others) is essential for spiritual growth [1]. This important spiritual dimension in the workplace occurs at the group level, where employees meet with others who work hand in hand with each other and develop strong communication between people [2,28]. 

Workplace spirituality has four levels: meaningful work, spirituality, compassion, and alignment of values [29]. Meaningful work can be defined as the pursuit of a higher purpose [30]. Spirituality is defined as the transcendent experience that a person has at work [31]. It can also be described as the level of sensible employees who get from their job [30]. Compassion is defined as the emotional connection between employees that motivates them to help others [32], and alignment of values can be defined as the harmony between personal beliefs and the norms of one’s work [33]. In other words, it is the psychological connection of employees with their workplace. The scriptures show that spirituality enables a person to benefit by increasing “happiness, peace, tranquility, satisfaction in work and commitment” [29]. Spirituality is also associated with increased creativity, honesty, and reliance on job verification between career growth and career goals [31]. Keeping in view all the work undertaken so far, we found a dearth of research on how different dimensions of workplace spirituality in academia are linked with psychological well-being and stress levels among university teachers.

### 1.1. Significance and Novelty of the Research

It is necessary to highlight the practice of workplace spirituality among university teachers because there are no formal studies on whether workplace spirituality is practiced in academia and what relation it has with the mental well-being of teachers. It is essential to consider the mental well-being and stress level of university teachers because a teaching activity cannot be conceived without the direct involvement of a teacher [33]. Even if we consider the progressing educationalists whose focus is mainly on the part and role of the learners, the role of a teacher, which is no less than that of a mentor, can never be ignored. Even for a learner, it will be impossible to groom and grow without a teacher’s direct or indirect involvement [34,35]. A teacher’s role is of utmost importance regarding logic, understanding, thinking, or morality. Parmar et al. [1] highlight the fact that it is a teacher who enables a learner to put his brain at work, ask essential and relevant questions, write, and grasp the concept from the reading, work in teams, understand a common good, and ultimately link the conscious with conduct. To summarize, a teacher’s role is of great importance in facilitating the growth and grooming of society as a whole and individual, curtailing democracy and morality [17]. Having such expectations and responsibilities, they must experience stress and psychological difficulties. The practice of workplace spirituality creates a mentally healthy environment and strengths the person’s capacity to deal with such stress. If teachers are not provided with the right workplace environment, it will be difficult for them to perform their duties properly [36]. Therefore, it is necessary to identify whether workplace spirituality is being practiced in academia or not. Besides this, it is essential to understand how it improves the overall mental well-being of university teachers [37,38]. 

### 1.2. The Objectives of the Research

The literature on workplace spirituality has several limitations. Therefore, the research has several objectives, for instance, to establish the impact of workplace spirituality on university teachers’ psychological well-being. Workplace spirituality is a new dimension in global work culture, since employees’ collective needs and values shift. Therefore, spirituality at work becomes a significant factor at work [35]. Research studies have primarily been conducted on a person’s spiritual experiences at the job instead of how the different workplace spirituality dimensions impact individuals’ work attitude and mental well-being [17,39]. The main objective of this research study is to investigate the relationship of different dimensions of workplace spirituality, e.g., compassion, relationship with others at work, spiritual orientation, organizational value, and alignment of personal value, with psychological well-being and job stress [38,40]. There is a gap in the literature, as primary workplace spirituality-related studies have been conducted on business employees or doctors [17,37] but not on teachers. Additionally, we also analyzed the impact of job stress, environmental mastery, and personal growth as mediating variables between independent variables, such as workplace spirituality, compassion, relationship with others at work, spiritual orientation, organizational value, and alignment of personal value, and psychological well-being [34,38].

## 2. Review of Literature

### 2.1. Theories Underpinning

We have incorporated and integrated the following three theories into the research study.

#### 2.1.1. Workplace Spirituality and Person—Organizational Support Theory

Workplace spirituality has four levels: meaningful work, spiritual orientation, compassion, and values alignment [29]. Theory of organizational support states that Perceived Organizational Support is valued partly as it is up to the point of the critical need for support, approval, self-esteem, and self-affiliation. Additionally, it is a significant cause of comfort during times of dire need and stress [40]. It can therefore be concluded that supportive supervision coupled with good human resources generates higher perceived organizational support. This perception leads to a better sense of satisfaction between the employee and the job [1,41]. Perceived organizational support conveys to the employees that the organization is willing to care and provide the required additional support when needed by the employee and will also reward the improved performance. It eventually results in better performance, lesser absenteeism, and better employee well-being. It has also been found that if the perceived support from the organization is less, this affects the employee, and thus employee’s general well-being is concerned, and job stress will be immense [42]. Sooner or later, the employee looks for an early exit compared to the opposite situation whereby employees remain with the organization even in harsh times [12,21]. 

#### 2.1.2. Person Organization Fit Theory

The Person organization fit theory has two processes. First, the sense of fulfillment and the fulfillment of a psychological need reflects a coherent view of equality, which must be understood by the extent to which workers’ needs are catered by the organization [43]. It suggests that dissatisfaction is found when necessities are unmet and may result in excessive fulfillment, depending on demand. The second tradition of beliefs in the proper study of person–organization fit theory is the concept of industrial-organizational congruence [44]. It is an additional method of equality [45] in which organizational values satisfy the employee’s needs. Theoretically, social interactions affect behavior because people are more attracted to similar colleagues and ultimately develop a relationship with other colleagues at work. Similarity helps communicate, ensures selections, and strengthens social identity [46].

#### 2.1.3. Ryff and Keyes’s (1995) Model of the Psychological Well-Being 

This theory is different from previous models. There are many forms of well-being, not only happiness or positive emotions. Good health incorporates various aspects of well-being instead of focusing on one or two aspects of health. Ryff and Keyes [47] classified well-being into the following six types:

##### Self-Acceptance

The person with high self-acceptance does not regret the past but feels satisfied. This individual believes positively about himself and acknowledges and accepts the multiple dimensions of his personality, which entail both positive and negative qualities [48]. Meanwhile, a person with low self-acceptance has a negative perception of himself, feels discontent, and always has regret about the past [44].

##### Personal Growth

An individual with solid personal growth is always growth and expression oriented. They have a drive and inner urge for continuous development. This person is more open to new and adventurous experiences [49]. While the individual with weak personal growth is more sluggish and has no drive or motivation towards personal development [1]. 

##### Purpose in Life

Individuals with a vital purpose in life usually set specific goals and directions. They assume to have a more meaningful present and valuable past [17]. They believe in the purpose of life with aims and objectives for life. Meanwhile, people with a weak purpose do not find any meaning in life, they have no clear direction [50].

##### Positive Relations with Others

People with a strong desire to have positive relations with others are concerned about the greater good and welfare at a more significant level. They are affectionate, intimidated, and believe in mutually beneficial human relationships [43]. Meanwhile, people with weak desires feel no warmth, openness, or concern about others. They usually have troubled interpersonal relations and are unwilling to compromise to have the relations [21].

##### Environmental Mastery

Individuals with high environmental mastery believe they have greater control over external events and are usually flexible [12]. While individuals with low environmental mastery feel disconnected from their surroundings. They have limited or no view of the existing opportunities and believe they have no control over the external elements of the environment [51]. 

##### Autonomy

Individuals with high autonomy believe in self and inner regulation. They evaluate things from a personal perspective and, can resist the pressure of social circle and act in the ways determined by self [52]. Meanwhile, people with low autonomy solely rely on others for making significant decisions and give up on social pressure to act and think in a certain way [1].

### 2.2. Spirituality

There is no single overall accepted explanation of spirituality at work; literature has more than 70 definitions of spirituality, all defining some aspects of it. Spirituality at work is defined in numerous possible ways and it is also defined as one’s self-consciousness. It is a sense of work that motivates and energizes employees to work at their full potential [53,54,55]. The literature differentiates spirituality from institutionalized religion. Spirituality is characterized as personal and inclusive. It is not related to any religion, belief system, rituals, or religious practices [22,54]. 

### 2.3. Workplace Spirituality

Workplace spirituality is a new dimension in global work culture, since employees’ collective needs and values shift. Therefore, spirituality at work becomes a significant factor at work. However, the religiosity belongs to someone’s faith, and faith could also be defined in several meanings or dimensions, but the workplace spirituality specifically belongs to the workplace culture. Workplace spirituality is a phenomenon that relates to the inquiry of what is the relationship between spirituality and work at the organization. In a broader sense, we can conceptualize it as expressive and experienced spirituality while considering the workplace and work [44,53]. Theoretically, it is the combined effect of workplace and spirituality. Still, it can also provide a new perspective of how the work of an individual can lead to his general development as an employee instead of being an environment with elements of hostility. In literature, this term is also linked with an individual’s spiritual connection with their working environment [17,54]. This phenomenon has recently received considerable attention from researchers, a relatively new term [41]. Spirituality at the workplace has shown a great potential for being a hot topic to study as it has been termed “the latest paradigm of organizational science” [42] with “the spirituality movement” [55]. Since it is mainly connected with employee outcomes, this can be of interest for the practitioners resulting in a significant shift of focus as organizations are more into building most of this phenomenon [39].

Although the term is a more personal and individual philosophical construct, and the literary meaning agrees with those deep values, a feeling of wholeness and a sense of connectivity with the organization are linked to workplace spirituality. It involves an effort directed towards self and finding out the reason for an individual’s existence in life, building upon the connections, personal and professional, linked with the work and creating a consistent linkage of the relationship between coworkers and link the internal belief system with that of the organization [34,56]. In this context, the phenomenon can be defined as “the sense that workers have a personal life that improves and is improved by meaningful work” [33]. There are several terms, and most do not qualify to describe the same phenomenon: spirituality in business, workplace spirituality or vice versa, and organizational spirituality. There cannot be a universal acceptance of the definition of workplace spirituality. For interpreting the term of workplace spirituality through a method that does not include the generalization or nonconcrete theoretical approach. Muzaki and Anggraeni [31] offer another explanation of spirituality: “Workplace spirituality is a framework of values of an organization showed in the culture which promotes workers experience of transcendence through the work process, facilitating their sense of being connected to others in a way that provided feelings of completeness and joy.”

After reviewing the literature, convergence takes place majorly in four themes that are recurrent, traceable, and termed as unified agreed upon: (1) integrating oneself with workplace (a complete and confined approach towards self and workplace); (2) purposefulness in work (a complete and confined approach towards self and workplace); (3) wholeness of self (giving up the self for the sake of wholeness and greater good); and (4) increased capabilities and self-understanding at a workplace [1,30]. The above-discussed dimensions are linked and found to be similar in most of the studies undertaken so far and thus include the subsequent parsing [29,57]. Firstly, integrating oneself with the workplace represent a complete and confined approach towards the self and workplace, is termed and explained as an individual’s intention to indulge his overall existence with an individual’s work or organization or not to recheck the spiritual element at the start. Thus, “people come to the workplace with their entire self and seek to incorporate work into their existence” [28]. Meanwhile, treating the workers’ workplace must take into consideration the fact that they are whole personalities with physical, mental, emotional, and spiritual needs [27]. Secondly, the dimensional aspect of workplace spirituality also covers the elements that an individual intends for wholesomeness, holism, and being integrated, but this often creates confusion with the term work in its functional definition, let aside the phenomenon of the workplace. Self-actualization is the third dimension, and it can be explained as “a greater sense of connection to something other than the self” [25]. Although, it has been found that there is confusion related to the fact that “something greater than oneself” could be part of the self. Resultantly, what is more, remarkable than the self is the individual surpassing. Self-development and the growth of an individual’s inner self at the workplace is the fourth dimension of the phenomenon, which is strongly connected with the other three aspects of the phenomenon of spirituality at the workplace [24,56]. Apart from the factor that imposes the quality of dynamism adding to the construct of spirituality. The process of relating workplace self-integration relates to the purpose of work and surpassing one must take place at work if the individual working has the innate wish to be integrated and whole at work. In turn, the underlying companies should “provide opportunities for workers to experience more personal growth and development” [21]. It has been quite controversial and confusing to discuss the purpose and rightfulness of workplace spirituality and its underlying connection with the performance of an organization [23,57]. The research undertaken in the ever-growing field of workplace spirituality lacks consolidation and a systematic approach to relating the company’s performance and spirituality.

The literature is focused on contributing to the phenomenon of spirituality at the workplace by combining the three differing perspectives (i.e., human resources perspective, philosophical perspective, and interpersonal perspective) on the relationship between spirituality and workplace or organizations output [1,22]. To better understand the terms, we first tend to define the issue from the human resources perspective. Its understanding entails the good outcomes of spirituality at the self-level but is not limited to commitment, well-being, and morale. It can be defined as workplace spirituality (a) improves the employees’ morale, thus resulting in better well-being, productivity, commitment, and eventually productivity, and on the other side (b) lowering the level of burnout, work holism, and stress at work [14]. Secondly, if we consider existential and philosophical dimensions, which interrelate with the terms like self-actualization, thus adding more meaning and purposefulness to the tasks undertaken at work. Providing a more in-depth understanding of meaningfulness and purposefulness is of greater importance because this enables employees to perform more productively and create more creativity and productivity at work [54]. This approach incorporates the fact that including or involving spirituality at work will provide the manager and employee with a more in-depth sense and purpose at the workplace [15]. Thirdly, if we apprehend the topic from a communal and interpersonal aspect, that is more in line with the idea of creating a sense of belongingness, connection, and community [16]. The essence of this approach is focused on the relationships with others, whole dimensions, and the social perspective of spirituality. This approach strengthens the idea that spirituality must be incorporated to create a sense of togetherness and wholeness and increase loyalty, belongings, and attachments at the workplace [18,57].

### 2.4. Psychological Well-Being in the Workplace

Generally, well-being can entail employees’ emotional, psychological, and mental aspects [53]. According to the literature, workplace well-being is assumed to be affected by three general factors [55]. Out of these three factors, the work setting is one of the critical components. Perils, safety, and health hazards can result in a harmful work environment, which ultimately results in effective health and well-being of the employees. These safety and health measures will be positive [58,59]. Workplace stress badly affects universities faculty members’ well-being [35]. According to Ravikumar [60], the source of occupational stress includes factors, such as overload or even underload, long shifts or not favorable shifts, and the overall environment of the workplace. At times, the roles are not clear, and in a workplace this can lead to [57,60] conflict regarding roles and extent or level of responsibility of others. Relationships at work involve seniors, line members, and even those reporting to an individual. Overall career growth or development increase uncertainty, insecurity, and important reforms within an organization. Similarly, climate change involves a lack of adequate consultation, participation, drawbacks in communication, communication gaps, and an uncertain and unknown work environment. An individual’s cultural misfit involves managing the work and life balance, including social and personal life, i.e., family time that provides social and emotional attachment with a family through a quality time spending [21,61]. According to literature, psychological well-being causes a couple of connected aspects of the aftereffects of workplace well-being [17,62]. These effects, rather than having more direct consequences, include behavioral, psychological, and even physical aftereffects. The second set of aftereffects entails absenteeism, health insurance costs, and productivity, which directly affect the workplace. Eventually, the setting of effects at an individual level will also affect the organizational level [63]. Thus, based on previous literature, we have framed the following hypotheses:

**Hypothesis** **1.**
*There is a significant positive relationship between workplace spirituality and psychological well-being.*


**Hypothesis** **2.***There is a significant positive relationship between compassion and psychological well-being*.

**Hypothesis** **3.**
*There is a significant positive relationship between relationships with others at work and psychological well-being.*


**Hypothesis** **4.**
*There is a significant positive relationship between spiritual orientation and psychological well-being.*


**Hypothesis** **5.**
*There is a significant positive relationship between organizational values and psychological well-being.*


**Hypothesis** **6.**
*There is a significant positive relationship between the alignment of personal values and psychological well-being.*


### 2.5. Mental Well-Being and Job Stress as Mediating Factors among Teachers

Amongst the “High-Risk Professions”, teaching has consistently been ranked as one of the most stressful jobs [64]. The research findings support it; many studies compared the levels of stress teachers go through affect their mental health. Other working people in different occupations have shown more mental stress in the teachers, level of stress, mental distress, and burnout [65]. It is a globally agreed-upon phenomenon that work-related demands and stress on the academic’s part have increased, and the trend is projected to intensify in the future [37]. The role of teachers is changing. There has been a rise in third-party scrutiny, contractual hiring, concurrent restructuring, an increase in workload, pressure to hunt grants and write research papers, and ultimately the accountability of in-class teaching [35,66]. Tertiary educators are making great efforts to balance the state of the workplace [67,68]. According to the research, academic employees are happier with intrinsic rewards, e.g., variety, responsibility, and flexibility, compared to extrinsic rewards, e.g., promotions or a salary increase. Connection with the community as a shared space also works as a stress releaser. For instance, according to Grandi et al. [69], socially supporting relations are a better way to cope with stress because of acceptance and availability [70]. Several studies have been conducted on workplace spirituality and its link with satisfaction at work, turnover intention [40], and other factors. Some studies were carried out on the link between spirituality in the workplace and the mental well-being of employees in other professions, and they used stress as a significant mediator [71]. However, the research lacks an understanding of the connection of spirituality in academia with the teachers’ psychological well-being and stress levels [1]. Very few research studies have been found on the mediating relationship between workplace spirituality, job stress, and its effect on the mental well-being of the employees of higher educational institutes [56,59]. This research aims to fill that gap and identify the different relation dimensions of workplace spirituality with psychological well-being and the mediating effect of job stress among university teachers. Thus, we framed the following hypothesis regarding the mediation of job stress between workplace spirituality and the psychological well-being of universities teachers [60,62,63,71]:

**Hypothesis** **7a.**
*Job stress mediates the relationship between workplace spirituality and psychological well-being.*


### 2.6. Mediation of Environmental Mastery and Personal Growth

The uniqueness and novelty of this modified model also lie in the mediation of job stress, environmental mastery, and personal growth [68]. Previous literature demonstrated that job stress is an imperative and decisive factor in enhancing psychological well-being concerning workplace spirituality and psychological well-being. Ryff and Keyes’s [47] model presented spirituality, workplace spirituality, personal growth, and environmental mastery [37,64,72]. Several studies demonstrated that environmental mastery has a significant and positive influence on organizational values, and alignment of personal values as a mediator [71,72,73], and the psychological well-being of universities teachers [74,75,76,77]. Similarly, personal growth is also mediated between spiritual orientation and psychological well-being [65,66,67]. The following hypotheses are framed based on previous literature:

**Hypothesis** **8a.**
*Environmental mastery mediates the relationship between organizational values and psychological well-being.*


**Hypothesis** **8b.**
*Environmental mastery mediates the relationship between the alignment of personal values and psychological well-being.*


**Hypothesis** **9a.**
*Personal growth mediates the relationship between spiritual orientation and psychological well-being.*


### 2.7. Theoretical and Conceptual Framework

We have integrated Ryff and Keyes’s [47] model of the psychological well-being person–organization fit theory [43] with workplace spirituality and person—organizational support theory [41], as depicted in Figure 1. Thus, we have driven the following direct and indirect hypotheses and theoretical and conceptual frameworks based on previous literature [21,33,34,39,41,47,77,78,79,80,81,82,83,84]. 

## 3. Materials and Methods

### 3.1. Research Design & Sampling Strategy 

The study was cross-sectional and quantitative, and we used a deductive approach to carry out this research. The study was conducted on 873 participants including male and female teachers. Respondents were university teachers, including permanent, visiting, and adjunct faculty members. They were selected through purposive sampling from Pakistan’s private and government sector universities. 

### 3.2. Instrumentations & Measurement Scaling

The data was collected using demographics sheets and adapted items from previous studies, for instance, the workplace spirituality scales were extracted from Franco-Santos and Doherty [35], and Ashmos and Duchon [82], and the items of job stress were extracted from Gillespie [83]. The measurement scales of psychological well-being, personal growth, environmental mastery, and relationship with others at work were taken from previous study, for instance, Ryff and Keyes [47] and measurement scales of job stress were extracted from Franco-Santos and Doherty [35], and Gillespie [83]. Measurement scaling of spiritual orientation, compassion, and alignment of values was taken from Pradhan [29], and items of organizational values were extracted from Schneider [84]. We had administered all the scales to all the participants. Thus, the measurement scaling was used from the earlier mentioned adopted questionnaires.

### 3.3. Data Collection Method

Before conducting the study, informed consent was taken from the participants, and they could withdraw from the study anytime. The sample population was cascaded questionnaire along with the demographic sheet, which includes their information regarding their age, gender, work experience, years served, area of specialization, the university they serve, highest qualification degree, and department or faculty names. We distributed 1000 questionnaires to the targeted sampling frame in person and online through their personal emails. However, 880 respondents returned the filled questionnaire. In 880 filled questionnaires, we found that 873 respondents, including male and female teachers, had comprehensively filled the questionnaires. Therefore, the response rate was recorded at 87.30%, which is considered a good number [2,85,86].

### 3.4. Estimation Techniques

Data were analyzed using SPSS 26, AMOS 26, Excel, and Conditional process modeling software v2.16.3. Firstly, we analyze demographics of our respondents through SPSS 26. Secondly, we used descriptive statistics to examine the fundamental characteristics of the constructs, and evaluated the normality pattern of the data through skewness and kurtosis. For this purpose, the data was converted into z-scores, and the descriptive statistics were extracted, if it showed that standard deviation and skewness were between ±1.50, and kurtosis ranges were between ±3.0 then the data is following the normality pattern. Thirdly, we employed exploratory factor analysis for data condensing and data validation, in which we employed rotated component matrix, factor loading, Cronbach’s alpha, composite reliability, average variance extracted, Kaiser–Meyer–Olkin (KMO) and Bartlett’s analyses, and total variance explained techniques. Therefore, these tests validated constructs and items, and data might be condensed, besides that we also analyzed the accuracy and suitability of the data from these techniques. After examining the data suitability and constructs and items validation, we employed SEM-based multivariate approach to examine the measurement model and structural model. For this purpose, we used two techniques, such as exploratory factor analysis and confirmatory factor analysis. We employed exploratory factor analysis to validate the measurement model through factor loading, Cronbach’s alpha, composite reliability, and average variance extracted. Additionally, the confirmatory factor analysis and structural equation modeling confirmed the measurement model and structure model through fit indices. Confirmatory factor analysis (CFA) is a direct method of examining observed and unobserved variables. We have examined the hypothesized structural model through structural equation modeling (SEM), as demonstrated by the previous literature [87,88,89]. Additionally, the fit indices validate the hypothesized measurement model [87,88]. In the end we employed direct and indirect relationships analysis among the independent variables, mediating & moderating variables, and dependent variable. We examined the direct relationship between independent and dependent variables using structural equation modeling through conditional process modeling [90]. We further used conditional process modeling to examine the mediating effect of job stress, environmental mastery, and personal growth between exogenous and endogenous variables. The conditional process modeling confirmed the hypothesized structural model in SEM-based multivariate approach. We also derived the readings through conditional process modeling and plotted the moderation through graphical analysis with the help of Excel software.

## 4. Results

### 4.1. Demographic Analysis

We had considered faculty members of private and public sector universities in Pakistan. Initially, we circulated 1000 questionnaires, and we received 873 questionnaires complete in a comprehensive manner. Hence, the response rate was 87.30%, which was considered to be a good number [2,85,86,91]. In our total sample size, we obtained 490 (56.1%) responses from males and 383 (43.9%) responses from females. The complete data of demographic analyses are reported in Table 1.

### 4.2. Descriptive Analysis

The descriptive statistics demonstrated the fundamental hallmarks of variables, such as standard deviation, mean, kurtosis, and skewness. The findings of Table 2 exhibited that standard deviation and skewness were between ±1.50, and Kurtosis ranges were between ±3.0. Thus, the normality of the data had been confirmed, which was a prerequisite for employing structural equation modeling [90,91,92]. However, there were several tests to check the normality of the data, but the simpler method to gauge the normality through skewness and kurtosis [93,94]. The next step was to employ the SEM-based multivariate techniques. The complete results of descriptive statistics are presented in Table 2.

### 4.3. Exploratory Factor Analysis—EFA

Table 3 exhibited that Cronbach’s alpha (CA) and composite reliabilities (CR) validated the reliabilities of constructs because values were higher than 0.70. Similarly, factor loading (FL) ranged between 0.75 and 0.95, which confirmed the convergent validities of an outer model, as the factor loading for each item should be higher than 0.70 [88,90]. The average variance extracted (AVE) met the condition of discriminant validities of construct because values are more significant than 0.50 [89]. Additionally, we computed KMO and Bartlett’s values, which further confirmed the suitability and validation of the data. Thus, in this way, the reliability and validity of all the considered items and constructs were achieved, and our hypothesized measurement model was proven and validated [90,91].

### 4.4. Kaiser-Meyer-Olkin (KMO) & Bartlett’s Analysis

The accuracy and reliability of the data were confirmed through Bartlett’s Sphericity and KMO analyses. According to Hair et al. [82], the value KMO analysis was 0.716, which ranged between 0.70–0.79 that considered a sufficient number. Similarly, Table 4 exhibited that the values of Bartlett’s sphericity analysis exhibit *p* < 0.05 at the 5% confidence interval level with 30,853.871 approximation of Chi-square and 946 degrees of freedom, which showed a significant correlation amongst the constructs. The validation and sampling adequacy were thus achieved [90]. Now, we can move forward with an advanced SEM-based multivariate approach [92].

### 4.5. Total Variance Explained

We also employed the total variance explained technique. The findings of Table 5 show a cumulative variance of considered ten components (variables) that showed 64.972 (64.97%) against the cut-off value of 0.5 or 50%. Additionally, we extracted the eigenvalues of every (component) construct, which were greater than one. Thus, we finally concluded the reliability and suitability of the data, which was also a prerequisite for the SEM-based multivariate approach [88].

### 4.6. Confirmatory Factor Analysis—CFA

The findings of Table 3 exhibit that our derived hypothesized measurement model, we considered six independent variables, for instance, workplace spirituality had five items. However, compassion, relationship with others at work, spiritual orientation, organizational value, and alignment of personal values had four items each construct. Moreover, we incorporated three mediating variables in our hypothesized measurement model, such as job stress and environmental mastery, which had four items each. However, personal growth has five items, the hypothesized measurement model had one dependent variable, and psychological well-being had five items. Thus, in this way, our considered hypothesized measurement model had ten constructs with forty-four items. The findings of Table 6 exhibit the fit-indices values for the measurement model, where the values of fit indices demonstrated that all the readings are within the threshold range [88,92]. Hence, it is concluded that our hypothesized measurement model was validated for the universities’ teachers’ psychological well-being.

### 4.7. Structure Equation Modeling—SEM

The findings of Table 3 demonstrate that the considered hypothesized structural model comprised six independent variables, e.g., workplace spirituality had five items. However, compassion, relationship with others at work, spiritual orientation, organizational value, and alignment of personal values had four items each construct. Moreover, we incorporated three mediating variables in our hypothesized structural model, such as job stress and environmental mastery, which had four items each. However, personal growth had five items, the hypothesized structural model had one dependent variable, and the psychological well-being has five items. Thus, in this way, our considered hypothesized measurement model had ten constructs with forty-four items. The findings of Table 6 exhibit the fit-indices values for the structural model that were within the threshold range [88,92]. Thus, it is concluded that our hypothesized structural model was validated for the universities’ teachers’ psychological well-being.

### 4.8. Hypothesized Direct Relationship

Table 7 exhibited that the independent variables, such as workplace spirituality, compassion, relationship with others at work, spiritual orientation, organizational value, and alignment of personal values, had a significant and positive relationship with psychological well-being. Thus, the hypotheses H1, H2, H3, H4, H5, and H6 were accepted because T-distribution values and corresponding probabilities are within the threshold limits (T > ±1.96 & *p* < 0.05), and it is finally concluded that universities’ teachers perceived psychological well-being (outcome variable) due to the independent variables, for instance, workplace spirituality, compassion, relationship with others at work, spiritual orientation, organizational value, alignment of personal values. The individual regressor alignment of personal values (0.4342) had the highest impact on teachers’ psychological well-being and was followed by the relationship with others at work (0.4015). However, compassion impacts 0.3926 on teachers’ well-being, and organizational value impacts 0.2560 on the universities’ teachers.

### 4.9. Mediation Analyses

We used 5000 numbers of bootstraps for bias corrected bootstrap confidence interval for every mediation examination. Table 8 exhibits the results obtained by means of two mediation methods, for instance, the normal theory, and bootstrapping methods. The bootstrapping method is normally, a more robust techniques in which we employed 5000 bootstraps for a single rotation, which was proposed by Hayes and Rockwood [95]. The Bootstrapping was a unique statistical approach, which resamples a single dataset to generate several simulated samples. However, the normal theory method is an alternative method to measure the mediation in which the statistics are employed to examine covariance structure. Techniques are usually based on the conjecture of multivariate normality. The findings of the normal theory method demonstrated that job stress significantly and negatively influenced workplace spirituality and psychological well-being because the effect of job stress is −0.0474 between exogenous and endogenous variables. Therefore, the findings reflected that job stress should be reduced to get optimal performance from university teachers, and job stress damages the psychological well-being of a university teacher. Similarly, environmental mastery had a significant and positive impact on the relationship of organizational value, alignment of personal values, and psychological well-being. Thus, the conducive working environment of an organization substantiated the psychological well-being in terms of organizational values and alignment of personal values. Finally, the mediation of personal growth also enhanced the psychological well-being of a university teacher with spiritual orientation (Z > ±1.96 & *p* < 0.05). Similar results were obtained from the bootstrapping method because zero did not lie between Boot LLCI (lower limit confidence interval) and Boot ULCI (upper limit confidence interval) [95]. Thus, it was also validated that the hypotheses H7a, H8a, H8b, and H9a were accepted, and it is concluded that job stress, environmental mastery, and personal growth had a significant impact as mediating variables between exogenous and endogenous variables [71].

## 5. Discussions

The current study aims to understand the relationship of workplace spirituality with the different dimensions of psychological well-being along with job stress, environmental mastery, and personal growth as mediating variables. The findings of a direct relationship exhibited that the independent variables, such as workplace spirituality, compassion, relationship with others at work, spiritual orientation, organizational value, and alignment of personal values, have a significant and positive relationship with psychological well-being. The findings of this research predicted a positive and robust impact of workplace spirituality on the overall psychological well-being of universities’ teachers, meaning that the higher the level of the practice of spirituality at work, the higher would be the level of psychological well-being [6,29,35]. Another study found that individuals who experience high workplace spirituality also have high self-esteem and experience a higher level of psychological well-being [9,27]. The practice of workplace spirituality increases self-esteem, which results in higher psychological security and satisfaction, which in return enhances psychological well-being [1,5,10]. Another study found that psychological well-being is predicted by workplace spirituality [7]. In the current study, compassion significantly and positively impacts psychological well-being. The previous literature also demonstrated similar results [11,13,17,57].

Similarly, the study’s findings exhibited that spiritual orientation has a significant and positive relationship with the psychological well-being of university teachers. The previous studies also demonstrated a similar outcome [12,14,15,35]. Spiritual orientation is the attitude, belief system, and knowledge that helps individuals in deriving meaning and purpose from life and work specifically, whereas personal growth is one of the dimensions of the psychological well-being model, which includes the development of skills and potential [16,18,57]. The hypothesis predicted a positive and significant relationship between the two: if the workplace has a spiritual orientation, the employees will have a higher level of psychological well-being [23,25]. This research was conducted with university teachers. It can be said that in academia, there is a lack of understanding of the importance of the well-being of the teachers, and there is some level of importance placed on physical well-being. For example, if a teacher is unwell physiologically, he/she can get leave, but when it comes to mental well-being, it is not recognized fully, and the understanding of the link between spiritual orientation and personal growth and their importance is not fully identified [20,21,57]. These are the possible reasons that teachers’ psychological well-being is compromised, and they are not mentally fit enough to have personal growth even in the presence of high spiritual orientation at the workplace [27,28,30].

The findings also predicted a positive and significant association between the alignment of personal & organizational values and the psychological well-being of university teachers. Previous literature also demonstrated consistent results [28,29,31]. It can be assumed that when an employee has his values aligned, he can understand and thus control crises more effectively [32,33]. The study conducted on workplace spirituality and turnover intentions among doctors suggests that when organizational values are aligned with personal values, employees can maintain harmony between work and life, which lowers the chances of turnover intentions [34,41]. Likewise, in the present study, teachers could control their surroundings and develop mastery. However, due to low levels of workplace spirituality and a higher stress level, they could not develop mastery over their environment, as the higher stress level affected their psychological well-being [34,35]. The findings further exhibited the positive and significant relationship relations with others at work and psychological well-being. Previous studies show that compassion in the workplace promotes better relations among employees. It improves the habits, e.g., recognizing and appreciating colleagues sincerely and working for the betterment of the organization. The practice of compassion at the workplace promotes engagement, dedication, empathy, cooperation, and kindness at the workplace [39,40,55].

The findings of mediation analysis of job stress in a relationship between workplace spirituality and psychological well-being were significantly negative [51,52,96], since it negatively mediates the relation between workplace spirituality and psychological well-being, meaning that with a high level of workplace spirituality, the job stress level will be low, then the psychological well-being level will be high, and vice versa [44,50,71]. Therefore, the findings reflect that job stress should be reduced to achieve optimal performance from university teachers, and job stress damages the psychological well-being of a university teacher [38,52,59]. The relationship between these two variables is a significant negative mediate relationship, which explains that the higher the stress level lower would be the psychological well-being [2,54]. The mean of job stress in the current study is more significant, which means the level of job stress is high in participants, even in the presence of moderate workplace spirituality due to the presence of high job stress levels and employee level of psychological well-being. The correlation of its dimensions with the different dimensions of workplace spirituality appears to be low [58,60]. It can also be said that due to the low level of workplace spirituality among university teachers, the stress levels were high, and therefore the level of psychological well-being was lower [37,66,68,70]. Similarly, environmental mastery has a significant and positive impact on the relationship of organizational value, alignment of personal values, and psychological well-being. Previous literature also substantiated that the conducive working environment of an organization alleviated psychological well-being in terms of organizational value and alignment of personal values [59,76,97]. Finally, the mediation of personal growth also enhances the psychological well-being of a university teacher with spiritual orientation [71,96,97].

## 6. Conclusions

Teaching is the most critical profession in an educated society. With teaching comes excellent responsibilities and distress. It is a lifelong learning process. Day in and out, there are new research findings and discoveries, science is proliferating, and the diffusion of technology has never been at such a pace. To effectively teach, a teacher these days must be familiar with the latest developments, research, discoveries, technologies, trends, and changes. With all such professional responsibilities, it is well understood that teachers are under great psychological stress, performance pressure, and other workplace distress. These challenges and responsibilities are exhausting, which can affect their well-being in challenging modern-day times. It is one of the significant challenges for educational institutes to identify ways to reduce or decrease job stress and improve psychological health and, therefore, better-quality education. The findings of a direct relationship exhibited that the independent variables, such as workplace spirituality, compassion, relationship with others at work, spiritual orientation, organizational value, and alignment of personal values, have a significant and positive relationship with psychological well-being. Teachers’ psychological well-being can be assured with the right workplace environment. Currently, with the rise of new challenges, teachers’ job is more complex. They must adapt to new ways of teaching and ensure that the quality of education is not compromised. Due to the increasing complexity and responsibilities, educational institutes must research positive workplace environment approaches such as workplace spirituality. It can help reduce stress and improve the teachers’ overall mental well-being. The mediation analyses demonstrated that job stress significantly and negatively influences workplace spirituality and psychological well-being. Therefore, the findings reflect that job stress should be reduced to get optimal performance from university teachers, and job stress damages the psychological well-being of a university teacher. Similarly, environmental mastery has a significant and positive impact on the relationship of organizational value, alignment of personal values, and psychological well-being. Thus, the conducive working environment of an organization substantiated the psychological well-being in terms of organizational values and alignment of personal values. Finally, the mediation of personal growth also enhances the psychological well-being of a university teacher with a spiritual orientation. Therefore, it is concluded that overall stress at the workplace is highly affecting the mental state and health of teachers, which affects their output as a teacher to students and their long-term mental and physical health. It is highly recommended that top-tier managers and bosses realize this fact and try to inculcate spirituality and other modern means of reducing job stress and promoting psychological well-being.

### 6.1. Theoretical and Practical Implications

The findings of this research provided a comprehensive conceptual framework to measure the universities’ teachers’ psychological well-being. Therefore, future researchers may replicate the similar model in other industries and other regions of the world since teaching is a profession that involves the highest possible levels of interaction with other people. It does not affect the mental state of the teachers but the students, who are the future of any country. It has been highly recommended for the top management to understand the importance of workplace spirituality. It is essential to understand that since the level of interactions is also high, the job stress that a teacher might feel may be further transmitted to students in terms of harsh behavior or strict checking and marking. Therefore, it must be realized that a healthy mental state will improve a teacher’s performance. If employees practice spirituality in the workplace, and as a result, there would be lesser job stress and more psychological well-being. The management must take appropriate steps to increase workplace spirituality. If higher educational institutes take any initiative to bring positive change, such initiatives are always trendsetters that later can be imparted to industry. It will also help bridge the industry-academia gap. Future research can also be conducted to understand the reason behind the low level of workplace spirituality and how it can be improved.

### 6.2. Limitations and Areas of Future Studies

This study has shown a statistically robust and significant relation between the direct and indirect association and the variables studied. Likewise, per the experts’ recommendations, the sample size was adequate to build upon the moderate to strong relationships amongst the variables. It has been evident that if the sample size is large, we tend to have more homogeneity, which enables us to have a more accurate prediction value of workplace spirituality for psychological well-being with job stress as a mediating variable. Getting the consent to participate from a particular target group is always tricky, and this is one of the limitations as this might affect the principle of generalization and end up misrepresenting the sample. Another limitation of the study was that we could not keep a balance or control the gender. Although females are fewer in number in terms of employment and higher positions, we still had more female participants. Therefore, it is recommended that future researchers add more female respondents for robust and generalizable outcomes. Moreover, as in the literature, these terms are still in the grooming phase in Pakistan. Therefore, most people and organizations have just started getting familiar with the subject discussed. Over time, once the concepts are fully mature and developed, the results might vary. The undertaken study has addressed the private and public sector universities’ professors. Therefore, it is recommended to the future researchers to replicate their studies in manufacturing and services sectors’ employees. The imperative limitation is not to gauge the cause and effect between the variables. Thus, it is recommended that future researchers conduct their studies by employing cause and effect statistical models [87].

## Figures and Tables

**Figure 1 ijerph-19-11244-f001:**
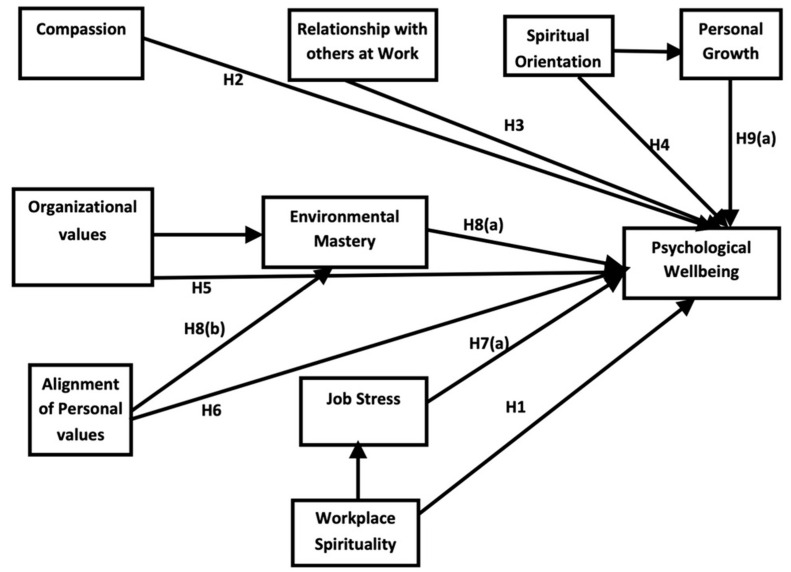
Theoretical and conceptual framework of the study. Source: Previous literature [21,29,33,34,39,41,44,47,77,78,79,80,81,82,83,84].

**Table 1 ijerph-19-11244-t001:** Demographic analyses.

Demographics		Frequency	Percent
Gender	Male	490	56.1%
	Female	383	43.9%
Marital Status	Single	504	57.7%
	Married	343	39.3%
	Divorced	26	3.0%
Age (In Years)	25–33	311	35.6%
	33–41	196	22.5%
	41–49	119	13.6%
	49–57	138	15.8%
	More than 57	109	12.5%
Education	Master degree	379	43.4%
	M.Phil. degree	279	32.0%
	Ph.D. degree	138	15.8%
	Post-doctorate	77	8.8%
Experience (In Years)	1–5	232	26.6%
	5–10	263	30.1%
	10–15	122	14.0%
	15–20	117	13.4%
	More than 20	139	15.9%
Income (In PKR 000)	45–75	150	17.2%
	75–110	386	44.2%
	110–145	179	20.5%
	145–180	97	11.1%
	More than 180	61	7.0%
Total—N		873	

**Table 2 ijerph-19-11244-t002:** Descriptive analyses.

Variables	N	Mean	Std. Deviation	Skewness		Kurtosis	
	Statistic	Statistic	Statistic	Statistic	Std. Error	Statistic	Std. Error
Psychological Well-being	873	3.7892	1.08324	−0.892	0.083	0.263	0.165
Workplace Spirituality	873	3.7640	1.07909	−0.821	0.083	0.184	0.165
Compassion	873	3.9290	1.10856	−0.953	0.083	0.313	0.165
Relationship with others at work	873	3.8396	1.06096	−0.930	0.083	0.515	0.165
Spiritual Orientation	873	3.7881	1.06648	−0.804	0.083	0.274	0.165
Organizational values	873	3.8202	1.04857	−0.934	0.083	0.583	0.165
Alignment of Personal values	873	3.9164	1.10508	−0.943	0.083	0.313	0.165
Job Stress	873	3.8625	1.07169	−0.935	0.083	0.472	0.165
Environmental Mastery	873	3.9278	1.10796	−0.952	0.083	0.315	0.165
Personal Growth	873	3.9003	1.09331	−0.942	0.083	0.373	0.165
Valid N (listwise)	873						

Note: standard error (SE) is the standard devistaion of its sampling distribution.

**Table 3 ijerph-19-11244-t003:** Measurement model.

Factors	Items	FL	CA	CR	AVE
Psychological Well-being	PW1	0.935	0.921	0.898	0.849
	PW2	0.879			
	PW3	0.937			
	PW4	0.927			
	PW5	0.901			
	PW6	0.948			
Workplace Spirituality	WS1	0.928	0.923	0.859	0.854
	WS2	0.906			
	WS3	0.928			
	WS4	0.930			
	WS5	0.927			
Compassion	COM1	0.943	0.924	0.959	0.854
	COM2	0.926			
	COM3	0.903			
	COM4	0.925			
Relationship with others at work	ROW1	0.931	0.908	0.950	0.826
	ROW2	0.904			
	ROW3	0.947			
	ROW4	0.852			
Spiritual Orientation	SO1	0.940	0.939	0.968	0.883
	SO2	0.943			
	SO3	0.925			
	SO4	0.951			
Organizational Values	OV1	0.950	0.926	0.960	0.859
	OV2	0.921			
	OV3	0.923			
	OV4	0.913			
Alignment of Personal values	APV1	0.927	0.925	0.959	0.856
	APV2	0.931			
	APV3	0.907			
	APV4	0.936			
Job Stress	JS1	0.931	0.916	0.954	0.840
	JS2	0.904			
	JS3	0.935			
	JS4	0.896			
Environmental Mastery	EM1	0.933	0.901	0.945	0.812
	EM2	0.934			
	EM3	0.943			
	EM4	0.856			
Personal Growth	PG1	0.876	0.905	0.850	0.820
	PG2	0.905			
	PG3	0.927			
	PG4	0.897			
	PG5	0.922			

Extraction Method: Principal Component Analysis; Rotation Method: Varimax with Kaiser Normalization. Note: FL = Factor loading; CA = Cronbach’s alpha; CR = Composite reliability; AVE = Average variance extracted; WS = Workplace Spirituality; COM = Compassion; ROW = Relationship with others at work; SO = Spiritual Orientation; OV = Organizational value; APV = Alignment of personal value; Mediating variables: JS = Job Stress; EM = Environmental mastery; PG = Personal growth; Dependent variables: PW = Psychological Well-being.

**Table 4 ijerph-19-11244-t004:** KMO and Bartlett’s Tests.

Kaiser-Meyer-Olkin Measure of Sampling Adequacy		0.716
Bartlett’s Test of Sphericity	Approximation of Chi-Square	30,853.871
	Degree of Freedom	946
	Significance Vale (*p*-Value)	0.000

**Table 5 ijerph-19-11244-t005:** Total Variance Explained.

Component	Initial Eigenvalues			Extraction Sums of Squared Loadings			Rotation Sums of Squared Loadings		
	Total	% of Variance	Cumulative %	Total	% of Variance	Cumulative %	Total	% of Variance	Cumulative %
1	6.129	13.929	13.929	6.129	13.929	13.929	6.086	13.831	13.831
2	3.122	7.096	21.025	3.122	7.096	21.025	2.608	5.928	19.759
3	2.898	6.586	27.612	2.898	6.586	27.612	2.601	5.912	25.671
4	2.750	6.250	33.861	2.750	6.250	33.861	2.599	5.907	31.578
5	2.642	6.004	39.866	2.642	6.004	39.866	2.587	5.880	37.457
6	2.595	5.897	45.762	2.595	5.897	45.762	2.586	5.877	43.334
7	2.548	5.791	51.553	2.548	5.791	51.553	2.572	5.845	49.179
8	2.365	5.375	56.928	2.365	5.375	56.928	2.567	5.835	55.014
9	2.315	5.261	62.190	2.315	5.261	62.190	2.551	5.798	60.812
10	1.934	4.396	66.585	1.934	4.396	66.585	1.830	4.160	64.972

Note: Component: Variable or construct.

**Table 6 ijerph-19-11244-t006:** Fit-indices statistic.

The Goodness of Fit Measures	Absolute Fit Indices			Relative Fit Indices			Non-Centrality-Based Indices			Parsimonious Fit Indices	
	χ^2^/df	Probability	GFI	NFI	IFI	TLI	CFI	RMSEA	RNI	PCFI	PNFI
Measurement Model	2.39	0.0225	0.96	0.95	0.97	0.96	0.97	0.037	0.98	0.85	0.86
Structural Model	3.01	0.0245	0.95	0.94	0.96	0.97	0.96	0.038	0.97	0.84	0.85
Criterion (Threshold values)	<5.0	<0.05	>0.95	>0.90	>0.95	>0.95	>0.95	<0.05	>0.95	>0.75	>0.75

Note: TLI = Tucker-Lewis Index; χ^2^/df = Relative Chi-square; GFI = Goodness of Fit Index; RMSEA = Root mean squared error of approximation; CFI = Comparative fit index; NFI = Normed fixed index; IFI = Incremental fixed index; RNI= Relative Non-centrality Index; PNFI = Parsimony-adjusted normed fit index; PCFI = Parsimonious-adjusted fit index.

**Table 7 ijerph-19-11244-t007:** Hypothesized direct relationship.

Hypotheses	Independent Variables	Dependent Variable	Regression Paths	StandardizedRegression Weights (β)	SE	T	*p*	Decision
H1:	Workplace Spirituality	Psychological Well-being	WS † → PW	0.1150	0.0257	6.02	0.0000	Supported
H2:	Compassion	Psychological Well-being	COM † → PW	0.3926	0.0308	12.75	0.0000	Supported
H3:	Relationship with others at work	Psychological Well-being	ROW † → PW	0.4015	0.0292	13.76	0.0000	Supported
H4:	Spiritual Orientation	Psychological Well-being	SO † → PW	0.1604	0.0335	7.94	0.0000	Supported
H5:	Organizational values	Psychological Well-being	OV † → PW	0.2560	0.0260	9.83	0.0000	Supported
H6:	Alignment of personal values	Psychological Well-being	APV † → PW	0.4342	0.0223	19.50	0.0000	Supported

Note: SE = Standard error; T = T-distribution values; *p* = Probabilities; † = Predictor; WS = Workplace Spirituality; COM = Compassion; ROW = Relationship with others at work; SO = Spiritual Orientation; OV = Organizational value; APV = Alignment of personal value; Dependent variables: PW = Psychological Well-being.

**Table 8 ijerph-19-11244-t008:** Mediation Analyses.

Hypotheses	Mediation	Bootstrapping Method				Normal Theory Method				
		Indirect Effect	Boot SE	Boot LLCI	Boot ULCI	Indirect Effect	SE	Z*	*p**	Decision
H7a:	WS→JS→PW	−0.0474	0.0284	−0.1035	0.0081	0.0474	0.0230	−2.06	0.0391	Supported
H8a:	OV→EM→PW	0.1182	0.0220	0.0770	0.1609	0.1182	0.0225	5.25	0.0000	Supported
H8b:	APV→EM→PW	0.0673	0.0163	0.0358	0.1000	0.0673	0.0192	3.50	0.0005	Supported
H9a:	SO→PG→PW	0.6414	0.0296	0.5853	0.3759	0.6414	0.0314	20.40	0.0000	Supported

Note: LLCI = Lower limit confidence interval; ULCI = Upper limit confidence interval; Predictor: WS = Workplace Spirituality; OV = Organizational value; APV = Alignment of personal values; SO = Spiritual Orientation; Mediating variables: JS = Job Stress; EM = Environmental Mastery; PG = Personal growth; Dependent variables: PW = Psychological Well-being; Z = Normal distribution values; SE = Standard error; *p* = Probability; Z* and *p** is significant at 5% confidence interval.

## Data Availability

Data is provided in Appendix A.

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
