# Peer review of "Relationship between Different Dimensions of Workplace Spirituality and Psychological Well-Being: Measuring Mediation Analysis through Conditional Process Modeling"

_ijerph, 2022, doi:10.3390/ijerph191811244_

Round 1
Reviewer 1 Report
This manuscript presents a novel insight into workplace spirituality and other factors which impact, yet moderated, the psychological well-being of the 873 university teachers in Pakistan, either from public or private universities.
At first sight, the paper is well defined, the literature review is sound, and the model is of utmost fit. The hypotheses are well defined and supported in previous studies, and the sample seems enough for the paper's purposes. However, there are several issues that should be fixed prior to considering this paper por publication, namely:
1) There is a good description of the factors that allow the researchers to build the model, rooting it in previous studies according to a theory. However, this seems a little weaker concerning the mediating factors. The paper would be much clearer for readers if the authors clarified why the factors have been defined as "mediators" and not "moderators", since the mediation effect needs to be highlighted prior to resuming the analysis (see Little, Todd & Card, N. & Bovaird, James & Preacher, K. & Crandall, Chris. (2007). Structural equation modeling of mediation and moderation with contextual factors. Modeling Contextual Effects in Longitudinal Studies), for instance.
2) Although the sample limitations have been acknowledged in the final part of the paper, indeed it is a problem. The authors simply state that 87,30% is considered a good number (twice, in lines 469 and 481-482). More support for this statement, on which are based all the efforts to come, needs to be provided, apart of the Ahmed et al. reference.
3) ŠtreimikienÄ— and Ahmed are not the only reference that should support something of such importance that the normality of the data (lines 489-491). More than a deviation/skewness/kurtosis description, a more complex analysis shul be run on this aspect (e. g. Kolmogorov-Smirnov). If it does not work, then try to log transform them.
4) In line 492 seems that SEM is going to be introduced right away, but it is not presented until line 550. Introduce preliminary analysis at that point would be helpful for the reader to follow up on the course of action and not get lost.
5) In part 4.8 (Mediation analyses) more information about the settings of bootstrapping method should be provided (e.g. how many bootstraps?). likewise add more comprehensive information to the reader about the difference between Normal Theory and Bootstr. methods.
6) Some references are hardly linked to the effects of what they are supposed to illustrate. For instance, Fontinha et al. (line 616) do have not much to do with the practice of spirituality at work. Another example is Kinman & Johnson, which is only a special section and not a regular paper (it does not contain the study mentioned in line 616).
7) There are some missing papers that should be included is it is intended to present the whole state of the art concerning well-being of teachers, for instance Franco-Santos, M., & Doherty, N. (2017). Performance management andwell-being: A close look at the changing nature of the U. K. highereducation workplace.The International Journal of Human ResourceManagement, 28,2319 –2350.http://dx.doi.org/10.1080/09585192.2017.1334148
Essentially the paper is a good piece of research, but in my opinion, needs to be refined in order to be fully understandable and accurate.
Author Response
Reviewer: 1
At first sight, the paper is well defined, the literature review is sound, and the model is of utmost fit. The hypotheses are well defined and supported in previous studies, and the sample seems enough for the paper's purposes. However, there are several issues that should be fixed prior to considering this paper por publication, namely:
Comments to the Author
1) There is a good description of the factors that allow the researchers to build the model, rooting it in previous studies according to a theory. However, this seems a little weaker concerning the mediating factors. The paper would be much clearer for readers if the authors clarified why the factors have been defined as "mediators" and not "moderators", since the mediation effect needs to be highlighted prior to resuming the analysis (see Little, Todd & Card, N. & Bovaird, James & Preacher, K. & Crandall, Chris. (2007). Structural equation modeling of mediation and moderation with contextual factors. Modeling Contextual Effects in Longitudinal Studies), for instance.
Authors’ Response:
We have highlighted mediation effect prior to resuming the analysis on Page No. 4 section 1.2, on Page No. 8 & 9 section 2.5, and Page No.9 section 2.6 as suggested by the reviewer. Moreover, we have substantiated the mediation by the citation provided by the reviewer such as: Todd & Card, N. & Bovaird, James & Preacher, K. & Crandall, Chris. (2007). Structural equation modeling of mediation and moderation with contextual factors. Modeling Contextual Effects in Longitudinal Studies). The citation is also added wherever it is needed. Thus, we have addressed the reviewer’s concern, and all the material is highlighted with RED color.
Comments to the Author
2) Although the sample limitations have been acknowledged in the final part of the paper, indeed it is a problem. The authors simply state that 87,30% is considered a good number (twice, in lines 469 and 481-482). More support for this statement, on which are based all the efforts to come, needs to be provided, apart of the Ahmed et al. reference.
Authors’ Response:
We have provided the justifications of 87.30% is a good number, and also substantiated with several citations on Page No. 11, section 3.3, and again on Page No. 11 section 4.1 as suggested by the Reviewer, and highlighted with RED color.
Comments to the Author
3) ŠtreimikienÄ— and Ahmed are not the only reference that should support something of such importance that the normality of the data (lines 489-491). More than a deviation/skewness/kurtosis description, a more complex analysis shul be run on this aspect (e. g. Kolmogorov-Smirnov). If it does not work, then try to log transform them.
Authors’ Response:
We have further provided the details of normality of the data through standard deviation, Skewness and Kurtosis, and substantiated with several citations on Page No. 12 section 4.2 as pointed out by the Reviewer, and highlighted with RED color.
Comments to the Author
4) In line 492 seems that SEM is going to be introduced right away, but it is not presented until line 550. Introduce preliminary analysis at that point would be helpful for the reader to follow up on the course of action and not get lost.
Author Response:
We have provided the details of all the used statistical techniques one-by-one, we have elaborated the preliminary analyses techniques before explaining SEM-based multivariate approach on Page No. 11, section 3.4 as recommended by the Reviewer, and highlighted with RED color.
Comments to the Author
5) In part 4.8 (Mediation analyses) more information about the settings of bootstrapping method should be provided (e.g. how many bootstraps?). likewise add more comprehensive information to the reader about the difference between Normal Theory and Bootstr. methods.
Author Response:
We have provided the comprehensive information of Bootstrapping method, and also differentiate the Bootstrapping method and Normal theory method on Page No. 16, section 4.9 as suggested by the Reviewer, and highlighted with RED color.
Comments to the Author
6) Some references are hardly linked to the effects of what they are supposed to illustrate. For instance, Fontinha et al. (line 616) do have not much to do with the practice of spirituality at work. Another example is Kinman & Johnson, which is only a special section and not a regular paper (it does not contain the study mentioned in line 616).
Author Response:
We have replaced more relevant references on the suggested material (line 616 & others), as recommended by the Reviewer, and highlighted them with RED color.
Comments to the Author
7) There are some missing papers that should be included it is intended to present the whole state of the art concerning the well-being of teachers, for instance Franco-Santos, M., & Doherty, N. (2017). Performance management and well being: A close look at the changing nature of the U. K. highereducation workplace.The International Journal of Human ResourceManagement, 28,2319 –2350.http://dx.doi.org/10.1080/09585192.2017.1334148
Author Response:
We have incorporated the missing and more relevant papers in our research article in “introduction”, Review of Literature”, and “Discussion” pertaining to teachers’ well-being including the suggested paper by the Reviewer, and highlighted with RED color in the citations that are in texts and references.

Reviewer 2 Report
Dear authors, I would like to thank you for reviewing this manuscript and encourage you to continue this research in other contexts.
I hope that the following comments will be useful to you and help to improve the quality of your work:
Title: In the title, include the context of the research: Relationship between different Dimensions of Workplace Spirituality and Psychological Wellbeing of University Teachers: Measuring Mediation analysis through Conditional Process Modeling
Line 27: In the sentence “…well-being was examined—finally,…”, substitute the hyphen (-) symbol with a dot (.). Also, capitalize the word “Finally”.
Lines 27-28: Include a verb in the sentence “Finally, mediation of personal growth between spiritual orientation and psychological well-being.”
Line 29: It is not clear if the sample of 873 participants corresponds to the sum of males and females, or 873 participants were males and 853 were females. Please specify.
Line 89: You write “A person’s…(singular)…their (plural). Please review and correct if necessary.
Lines 105-106: Review and correct the following sentence if necessary “The level of spirit one 105 defines a person's expression or spiritual knowledge at work puts at their job.”
Line 115: What do you mean by “it”? It seems you refer to workplace spirituality. If so, review and correct the sentence if necessary.
Line 135-137: It is not clear if spirituality and spiritual standing are the same. Please specify.
Line 149: Section 2.1 Significance and novelty of the research must be numbered 1.1, not 2.1.
Line 172: Section 2.2 The objectives of the research must be numbered 1.2, not 2.2.
Line 189-192: In section 2.1 you mention you incorporate three theories. Then, sections 2.2, 2.3, and 2.4 (where it is supposed you mention the three theories) must be numbered as subsections of section 2.1 (i.e. 2.1.1 Workplace spirituality and person – Organizational support theory, 2.1.2 Person organization fit theory, and 2.1.3 Ryff and Keyes's (1995) model of the Psychological Well-being).
Lines 209 and 221. Sections 2.3 Person organization fit theory and 2.3 Ryff and Keyes's (1995) model of the Psychological Well-being are numbered in the same way (2.3). Review and correct them.
Line 257: In the sentence “…personal perspective ad can resist…”, I think you want to say “and” instead of “ad”. Review and correct if necessary.
Lines 287-288: It seems that the following sentence is not complete. “To interpret the term by an approach that does not include the generalization or abstract theoretical approach.”
Review and correct it if necessary.
Line 306: I think the parenthesis “)” must be removed.
Line 367: In the brackets [21.59] the dot (.) must be changed by a coma (,).
Line: In the sentence “According to Ravikumar [57], the source of occupational stress includes factors like;”, change the semicolon (;) by a colon (:).
Lines 362-365: The sentence ”Overall career growth or development; growing uncertainty and insecurity; structural reforms within an organization and climate change which entails lack of adequate consultation, participation, drawbacks in communication, communication gap, uncertain and unknown work environment” is not clear. A verb is necessary. Review and correct it.
Lines 365-366: The sentence “An individual's cultural misfit; managing the work and life balance, including social and personal life, which includes family time” is not clear, it needs a complement.
Lines 367-368: English must be improved. For instance, in the sentence “According to literature entails a couple of connected aspects of the aftereffects of workplace well-being”, a coma (,) must be added after “literature; and it is not clear who entails a couple of connected aspects.
Figure 1: Improve the aesthetic quality of the figure, do not cut rectangles with the arrow lines (as with Job stress). The arrow lines should not go into the rectangles (as with Alignment of personal values).
Lines 453-455: The sentence “The measurement scales of psychological well-being, personal growth, environmental mastery, and relationship with others at work [46] and measurement scales of job stress from Gillespie [79]” is meaningless, a verb is missing. Include a verb.
Section 3.2: Include the three questionnaires with the corresponding items and scales.
Line 465: Avoid using “etc.”. Instead of that, trite all the data you ask for, or include a Figure of the demographic sheet.
Section 3.4: You mention the parameters you used to validate the measurement model. However, you must be more specific. For example, what do you validate with factor loading? What do you validate with Cronbach’s alpha? And son with the other parameters. Moreover, you have to mention what are the acceptable values or ranges of acceptable values for each parameter to say that the measurement instrument was validated.
Line 475: Be sure to define the acronym KMO before using it.
Section 3.5: The information presented in this section does not correspond to the section of Materials and Methods. You are not explaining HOW you do the research. You are mentioning WHAT you obtained after you performed the demographic analysis. Then, the information presented in section 3.5 corresponds to the Results. If you want to include a section named 3.5 Demographic analysis, then you must explain HOW you do that demographic analysis.
Lines 488-493: The following two sentences correspond to the section of Materials and Methods since you are explaining HOW you did your research: “The data were converted into z-scores, and the descriptive statistics were extracted” and “The next step is to employ the SEM-based multivariate techniques”.
Table 2: In some columns, you have written Statistics, while in others it just says Statis-. Be consistent or clarify if there is a difference between the two words. In addition, it is necessary to clarify what Std. Er-
Lines 497-501: The following two sentences correspond to the section of Materials and Methods since you are explaining HOW you did your research: “The next step is to validate the measurement model; for this purpose, we used exploratory factor analysis. For the data condense and items & constructs validation, we employed a rotated component matrix in which we computed factor loadings of items, Cronbach’s alpha, composite reliabilities, and average variance extracted (AVE) of constructs.”
Table 3: Be sure to define the acronyms FL, CA, and CR before using them.
Table 4: Titles must not have a final dot (Kaiser-Meyer-Olkin Measure of Sampling Adequacy.). Explain the meaning of the acronyms Df and Sig. What about the values of Df, Sig. and Approx. Chi-Square? What do these values mean? Is Sig. corresponding to the p-value? It is not clear.
Table 5: What do you mean by “Component” in Table 5? It is not clear.
Lines 534-543: The text in these lines does not inform any result. It appears to belong to the Materials and Method section.
Lines 537-538: The sentence “However, compassion, relationship with others at work, spiritual orientation, organizational value, and alignment of personal values with four items each” must be improved. A verb is necessary.
Lines 545-546: What do the following acronyms mean GFI=0.96, CFI=0.97, RNI=0.98, IFI=0.97, NFI=0.95, TLI=0.96, PCFI=0.85, PNFI=0.86, and RMSEA=0.037.? And how can you affirm that all the fit indices demonstrate that all the readings are within the threshold range? It is not clear what is the threshold range for each index.
Lines 554-555 The sentence “However, compassion, relationship with others at work, spiritual orientation, organizational value, and alignment of personal values with four items each.” Must be improved. A verb is necessary.
Lines 551-561: The text in these lines does not inform any result. It appears to belong to the Materials and Method section.
Lines 561-562: What do the following acronyms mean GFI=0.96, CFI=0.97, RNI=0.98, IFI=0.97, NFI=0.95, TLI=0.96, PCFI=0.85, PNFI=0.86, and RMSEA=0.037.? And how can you affirm that all the fit indices demonstrate that all the readings are within the threshold range? It is not clear what is the threshold range for each index.
Lines 566-568: The following sentence corresponds to the section of Materials and Methods since you are explaining HOW you did your research: “We examined the direct relationship between independent and dependent variables using structural equation modeling through conditional process modeling (Ahmed et al., 567 2022b)”.
Table 6: Make sure that the acronyms SE, T, and P are defined. Does the capital P in the Table represent the same as the lower-case p in the text? Clarify that.
Lines 584-586: The following sentence corresponds to the section of Materials and Methods since you are explaining HOW you did your research: “We have used conditional process modeling to examine the mediating effect of job stress, environmental mastery, and personal growth between exogenous and endogenous variables.)”.
Lines 586-587: Table 7 does not exhibit the two mediation methods. Table 7 shows the results obtained by means of these two methods.
Lines 587-589: How can you substantiate that the findings of the Normal theory method demonstrated that job stress significantly and negatively influences workplace spirituality and psychological well-being?
Line 589: Be sure to define the acronyms LLCI and ULCI.
Table 7: Be sure to define the acronym SE. and the abbreviation Prob. Why Z and Prob. have * and **, respectively? Do you want to make an indication about this?
Lines 658-659: The following sentence is confusing “To test the mediation analysis, which predicted that job stress mediates the relationship between workplace spirituality and psychological well-being [50,51,86]”. Please read and correct it.
I finished reading the manuscript and I do not understand what spirituality is, and then, I do not have a clear idea of what workplace spirituality is. Please provide a clear and detailed definition of spirituality before defining workplace spirituality.
Sometimes you use wellbeing and other times well-being. Be constant and use only one.
In Materials and Method, you have to detailly explain HOW you do the research, and every material, instrument, and technique you use. For example, how and for what purpose do you apply the total variance explained technique, and what are the important values of this technique for decision making? And so on with each technique.
Be sure to write the Results section in the past tense.
Author Response
Reviewer: 2
Dear authors, I would like to thank you for reviewing this manuscript and encourage you to continue this research in other contexts. I hope that the following comments will be useful to you and help to improve the quality of your work:
Title: In the title, include the context of the research: Relationship between different Dimensions of Workplace Spirituality and Psychological Wellbeing of University Teachers: Measuring Mediation analysis through Conditional Process Modeling
Comments to the Author
1) Line 27: In the sentence “…well-being was examined—finally,…”, substitute the hyphen (-) symbol with a dot (.). Also, capitalize the word “Finally”.
Authors’ Response:
We have made correction on Page No. 1 in the suggested lines, and highlighted with RED color.
Comments to the Author
Lines 27-28: Include a verb in the sentence “Finally, mediation of personal growth between spiritual orientation and psychological well-being.”
Authors’ Response:
We have made correction on Page No. 1 in the suggested lines and included the verb, and highlighted with RED color.
Comments to the Author
Line 29: It is not clear if the sample of 873 participants corresponds to the sum of males and females, or 873 participants were males and 853 were females. Please specify.
Authors’ Response:
We have made correction on Page No. 1, and specify the sample pertaining to males and females as recommended by the Reviewer in the suggested lines, and highlighted with RED color.
Comments to the Author
Line 89: You write “A person’s…(singular)…their (plural). Please review and correct if necessary.
Authors’ Response:
We have made correction as suggested by the Reviewer on Page No. 2 in the suggested line, and highlighted with RED color.
Comments to the Author
Lines 105-106: Review and correct the following sentence if necessary “The level of spirit one 105 defines a person's expression or spiritual knowledge at work puts at their job.”
Authors’ Response:
We have corrected the wording of the sentence on Page No. 3 as suggested by the Reviewer, and highlighted with RED color.
Comments to the Author
2) Line 115: What do you mean by “it”? It seems you refer to workplace spirituality. If so, review and correct the sentence if necessary.
Authors’ Response:
We have made correction as suggested by the Reviewer on Page No. 3 in the suggested line, and highlighted with RED color.
Comments to the Author
Line 135-137: It is not clear if spirituality and spiritual standing are the same. Please specify.
Authors’ Response:
We have made correction as spirituality on Page No.3 as pointed out by the Reviewer in the suggested lines, and highlighted with RED color.
Comments to the Author
Line 149: Section 2.1 Significance and novelty of the research must be numbered 1.1, not 2.1.
Authors’ Response:
We have made corrections in numbers of sub-sections on Page No. 3 & 4 as pointed out by the Reviewer in the suggested line, and highlighted with RED color.
Comments to the Author
Line 172: Section 2.2 The objectives of the research must be numbered 1.2, not 2.2.
Authors’ Response:
We have made correction as in sub-section as pointed out by the Reviewer on Page No. 3 & 4 in the suggested line, and highlighted with RED color.
Comments to the Author
Line 189-192: In section 2.1 you mention you incorporate three theories. Then, sections 2.2, 2.3, and 2.4 (where it is supposed you mention the three theories) must be numbered as subsections of section 2.1 (i.e. 2.1.1 Workplace spirituality and person – Organizational support theory, 2.1.2 Person organization fit theory, and 2.1.3 Ryff and Keyes's (1995) model of the Psychological Well-being).
Authors’ Response:
We have made correction in sub-section as suggested by the Reviewer on Page No. 4, 5, 6, 8 & 9 in the suggested lines, and highlighted with RED color.
Comments to the Author
Lines 209 and 221. Sections 2.3 Person organization fit theory and 2.3 Ryff and Keyes's (1995) model of the Psychological Well-being are numbered in the same way (2.3). Review and correct them.
Authors’ Response:
We have made correction in sub-section as suggested by the Reviewer on Page No. 4, 5, 6, 8 & 9 in the suggested lines, and highlighted with RED color.
Comments to the Author
3) Line 257: In the sentence “…personal perspective ad can resist…”, I think you want to say “and” instead of “ad”. Review and correct if necessary.
Authors’ Response:
We have made correction of the wording “and” on Page No. 6 as suggested by the Reviewer in the suggested lines, and highlighted with RED color.
Comments to the Author
Lines 287-288: It seems that the following sentence is not complete. “To interpret the term by an approach that does not include the generalization or abstract theoretical approach.”
Review and correct it if necessary.
Authors’ Response:
We have made correction in the sentence on Page No. 6 as suggested by the Reviewer in the suggested lines, and highlighted with RED color.
Comments to the Author
Line 306: I think the parenthesis “)” must be removed.
Authors’ Response:
We have made correction in the sentence on Page No. 6 as suggested by the Reviewer in the suggested lines, and highlighted with RED color.
Comments to the Author
Line 367: In the brackets [21.59] the dot (.) must be changed by a coma (,).
Authors’ Response:
We have made correction in the citation on Page No. 8 as suggested by the Reviewer in the suggested line, and highlighted with RED color.
Comments to the Author
Line: In the sentence “According to Ravikumar [57], the source of occupational stress includes factors like;”, change the semicolon (;) by a colon (:).
Authors’ Response:
We have replaced semicolon (;) by a colon (:) in the sentence on Page No. 8 as suggested by the Reviewer in the suggested lines, and highlighted with RED color.
Comments to the Author
Lines 362-365: The sentence ”Overall career growth or development; growing uncertainty and insecurity; structural reforms within an organization and climate change which entails lack of adequate consultation, participation, drawbacks in communication, communication gap, uncertain and unknown work environment” is not clear. A verb is necessary. Review and correct it.
Authors’ Response:
We have made correction in the sentence and also added a verb on Page No. 8 as suggested by the Reviewer in the suggested lines, and highlighted with RED color.
Comments to the Author
4) Lines 365-366: The sentence “An individual's cultural misfit; managing the work and life balance, including social and personal life, which includes family time” is not clear, it needs a complement.
Authors’ Response:
We have made correction in the sentence on Page No. 8 as suggested by the Reviewer in the suggested lines, and highlighted with RED color.
Comments to the Author
Lines 367-368: English must be improved. For instance, in the sentence “According to literature entails a couple of connected aspects of the aftereffects of workplace well-being”, a coma (,) must be added after “literature; and it is not clear who entails a couple of connected aspects.
Authors’ Response:
We have made correction in the sentence on Page No. 8 as suggested by the Reviewer in the suggested lines, and highlighted with RED color.
Comments to the Author
Figure 1: Improve the aesthetic quality of the figure, do not cut rectangles with the arrow lines (as with Job stress). The arrow lines should not go into the rectangles (as with Alignment of personal values).
Authors’ Response:
We have made correction in the Figure 1 on Page No. 10 as suggested by the Reviewer in the suggested rectangles and arrows, and highlighted with RED color.
Comments to the Author
Lines 453-455: The sentence “The measurement scales of psychological well-being, personal growth, environmental mastery, and relationship with others at work [46] and measurement scales of job stress from Gillespie [79]” is meaningless, a verb is missing. Include a verb.
Authors’ Response:
We have made correction in the sentence, and also added a verb on Page No. 10, section 3.2 as suggested by the Reviewer in the suggested lines, and highlighted with RED color.
Comments to the Author
Section 3.2: Include the three questionnaires with the corresponding items and scales.
Authors’ Response:
We have made corrections in the sentence, and overall sub-section 3.2 on Page No. 10 as suggested by the Reviewer, and highlighted with RED color.
Comments to the Author
Line 465: Avoid using “etc.”. Instead of that, trite all the data you ask for, or include a Figure of the demographic sheet.
Authors’ Response:
We have made correction in the sentence and avoid etc., and provided all the information on Page No. 10, section 3.3 as suggested by the Reviewer in the suggested lines, and highlighted with RED color.
Comments to the Author
5) Section 3.4: You mention the parameters you used to validate the measurement model. However, you must be more specific. For example, what do you validate with factor loading? What do you validate with Cronbach’s alpha? And son with the other parameters. Moreover, you have to mention what are the acceptable values or ranges of acceptable values for each parameter to say that the measurement instrument was validated.
Authors’ Response:
We have provided all the suggested information on Page No. 11, section 3.4 as suggested by the Reviewer, and highlighted with RED color. We have also provided the acceptable values of all the parameters in results section, and highlighted with RED color.
Comments to the Author
Line 475: Be sure to define the acronym KMO before using it.
Authors’ Response:
We have defined the acronym KMO on Page No. 11, section 3.4 as suggested by the Reviewer, and highlighted with RED color.
Comments to the Author
Section 3.5: The information presented in this section does not correspond to the section of Materials and Methods. You are not explaining HOW you do the research. You are mentioning WHAT you obtained after you performed the demographic analysis. Then, the information presented in section 3.5 corresponds to the Results. If you want to include a section named 3.5 Demographic analysis, then you must explain HOW you do that demographic analysis.
Authors’ Response:
We have shifted demographic analysis in the results section on Page No. 11, as section 4.1 as suggested by the Reviewer, and highlighted with RED color.
Comments to the Author
Lines 488-493: The following two sentences correspond to the section of Materials and Methods since you are explaining HOW you did your research: “The data were converted into z-scores, and the descriptive statistics were extracted” and “The next step is to employ the SEM-based multivariate techniques”.
Authors’ Response:
We have shifted the two sentences to the section of Material and Methods as pointed out by the Reviewer on Page No. 11, in section 3.4 as suggested by the Reviewer in the suggested lines, and highlighted with RED color.
Comments to the Author
Table 2: In some columns, you have written Statistics, while in others it just says Statis-. Be consistent or clarify if there is a difference between the two words. In addition, it is necessary to clarify what Std. Er-
Authors’ Response:
We have made corrections, actually they were hide, we have unhidden the text of columns of Table 2 on Page No. 12, as suggested by the Reviewer, and highlighted with RED color. We have also defined the Standard Error.
Comments to the Author
Lines 497-501: The following two sentences correspond to the section of Materials and Methods since you are explaining HOW you did your research: “The next step is to validate the measurement model; for this purpose, we used exploratory factor analysis. For the data condense and items & constructs validation, we employed a rotated component matrix in which we computed factor loadings of items, Cronbach’s alpha, composite reliabilities, and average variance extracted (AVE) of constructs.”
Author Response:
We have shifted the two sentences to the section of Material and Methods as pointed out by the Reviewer on Page No. 11, in section 3.4 as suggested by the Reviewer in the suggested lines, and highlighted with RED color.
Comments to the Author
6) Table 3: Be sure to define the acronyms FL, CA, and CR before using them.
Author Response:
We have defined the acronyms FL, CA, CR, and AVE before using it on Page No. 13, in section 4.3 as suggested by the Reviewer in the suggested lines, and highlighted with RED color.
Comments to the Author
Table 4: Titles must not have a final dot (Kaiser-Meyer-Olkin Measure of Sampling Adequacy.). Explain the meaning of the acronyms Df and Sig. What about the values of Df, Sig. and Approx. Chi-Square? What do these values mean? Is Sig. corresponding to the p-value? It is not clear.
Author Response:
We have omitted a dot (Kaiser-Meyer-Olkin Measure of Sampling Adequacy.), and also explained the Df, Sig. values on Page No. 14, in Table 4 as suggested by the Reviewer, and highlighted with RED color.
Comments to the Author
Table 5: What do you mean by “Component” in Table 5? It is not clear.
Author Response:
We have defined the components in the footer of Table 5 on Page No. 14 & 15, as suggested by the Reviewer, and highlighted with RED color.
Comments to the Author
Lines 534-543: The text in these lines does not inform any result. It appears to belong to the Materials and Method section.
Author Response:
We have shifted the sentences to the section of Material and Methods as pointed out by the Reviewer on Page No. 11, in section 3.4 as suggested by the Reviewer in the suggested lines, and highlighted with RED color.
Comments to the Author
Lines 537-538: The sentence “However, compassion, relationship with others at work, spiritual orientation, organizational value, and alignment of personal values with four items each” must be improved. A verb is necessary.
Authors’ Response:
We have made correction in the sentence, and also added a verb on Page No. 15, section 4.6 as suggested by the Reviewer in the suggested lines, and highlighted with RED color.
Comments to the Author
Lines 545-546: What do the following acronyms mean GFI=0.96, CFI=0.97, RNI=0.98, IFI=0.97, NFI=0.95, TLI=0.96, PCFI=0.85, PNFI=0.86, and RMSEA=0.037.? And how can you affirm that all the fit indices demonstrate that all the readings are within the threshold range? It is not clear what is the threshold range for each index.
Authors’ Response:
We have provided a comprehensive Table 6 for all the Fit-indices with measurement model and structural model values. Moreover, we have also provided the threshold values with the meanings of all the acronyms in the footer of Table 6 on Page No. 15, as section 4.6 as suggested by the Reviewer in the suggested lines, and highlighted with RED color.
Comments to the Author
7) Lines 554-555 The sentence “However, compassion, relationship with others at work, spiritual orientation, organizational value, and alignment of personal values with four items each.” Must be improved. A verb is necessary.
Authors’ Response:
We have made correction in the sentence, and also added a verb on Page No. 15, section 4.7 as suggested by the Reviewer in the suggested lines, and highlighted with RED color.
Comments to the Author
Lines 551-561: The text in these lines does not inform any result. It appears to belong to the Materials and Method section.
Author Response:
We have shifted the sentences to the section of Material and Methods as pointed out by the Reviewer on Page No. 11, in section 3.4 as suggested by the Reviewer in the suggested lines, and highlighted with RED color.
Comments to the Author
Lines 561-562: What do the following acronyms mean GFI=0.96, CFI=0.97, RNI=0.98, IFI=0.97, NFI=0.95, TLI=0.96, PCFI=0.85, PNFI=0.86, and RMSEA=0.037.? And how can you affirm that all the fit indices demonstrate that all the readings are within the threshold range? It is not clear what is the threshold range for each index.
Authors’ Response:
We have provided a comprehensive Table 6 for all the Fit-indices with measurement model and structural model values. Moreover, we have also provided the threshold values with the meanings of all the acronyms in the footer of Table 6 on Page No. 15, as section 4.7 as suggested by the Reviewer in the suggested lines, and highlighted with RED color.
Comments to the Author
Lines 566-568: The following sentence corresponds to the section of Materials and Methods since you are explaining HOW you did your research: “We examined the direct relationship between independent and dependent variables using structural equation modeling through conditional process modeling (Ahmed et al., 567 2022b)”.
Author Response:
We have shifted the sentences to the section of Material and Methods as pointed out by the Reviewer on Page No. 11, in section 3.4 as suggested by the Reviewer in the suggested lines, and highlighted with RED color.
Comments to the Author
Table 6: Make sure that the acronyms SE, T, and P are defined. Does the capital P in the Table represent the same as the lower-case p in the text? Clarify that.
Author Response:
We have defined the acronyms SE, T, and P, yes P & p is the same we have made uniformity now as pointed out by the Reviewer on Page No. 15 & 16, in section 4.8 & Table 7 as suggested by the Reviewer in the suggested lines, and highlighted with RED color.
Comments to the Author
Lines 584-586: The following sentence corresponds to the section of Materials and Methods since you are explaining HOW you did your research: “We have used conditional process modeling to examine the mediating effect of job stress, environmental mastery, and personal growth between exogenous and endogenous variables.)”.
Author Response:
We have shifted the sentences to the section of Material and Methods as pointed out by the Reviewer on Page No. 11, in section 3.4 as suggested by the Reviewer in the suggested lines, and highlighted with RED color.
Comments to the Author
Lines 586-587: Table 7 does not exhibit the two mediation methods. Table 7 shows the results obtained by means of these two methods.
Author Response:
We have made corrections in sentences of section 4.9, Table 7 (Now Table 8) on Page No. 16 & 17 as suggested by the Reviewer in the suggested lines, and highlighted with RED color.
Comments to the Author
Lines 587-589: How can you substantiate that the findings of the Normal theory method demonstrated that job stress significantly and negatively influences workplace spirituality and psychological well-being?
Author Response:
We have defined the reasoning regarding the pointed out sentence by the Reviewer made corrections in sentences of section 4.9, on Page No. 16 as suggested by the Reviewer in the suggested lines, and highlighted with RED color.
Comments to the Author
Line 589: Be sure to define the acronyms LLCI and ULCI.
Author Response:
We have defined the acronyms of LLCI and ULCI on Page No. 16 & 17 as suggested by the Reviewer in the suggested lines, and highlighted with RED color.
Comments to the Author
Table 7: Be sure to define the acronym SE. and the abbreviation Prob. Why Z and Prob. have * and **, respectively? Do you want to make an indication about this?
Author Response:
We have defined the acronyms of SE, and abbreviations of Prob**., Z*, and also made corrections for (*) and (**) on Page No. 17, Table 8, as suggested by the Reviewer in the suggested lines, and highlighted with RED color.
Comments to the Author
Lines 658-659: The following sentence is confusing “To test the mediation analysis, which predicted that job stress mediates the relationship between workplace spirituality and psychological well-being [50,51,86]”. Please read and correct it.
Author Response:
We have made correction in the sentence that is pointed out by the by the Reviewer in the suggested lines on Page No. 18, and highlighted with RED color.
Comments to the Author
I finished reading the manuscript and I do not understand what spirituality is, and then, I do not have a clear idea of what workplace spirituality is. Please provide a clear and detailed definition of spirituality before defining workplace spirituality.
Authors’ Response:
We have comprehensively discussed and differentiated the spirituality, spirituality at work, and religiosity for the general readers on Page No. 6, section 2.1, Page No. 6, 7 & 8, section 2.2 as recommended by the Reviewer, and highlighted with RED color.
Comments to the Author
Sometimes you use wellbeing and other times well-being. Be constant and use only one.
Authors’ Response:
We have made uniformity as well-being throughout the paper as recommended by the Reviewer, and highlighted with RED color.
Comments to the Author
In Materials and Method, you have to detailly explain HOW you do the research, and every material, instrument, and technique you use. For example, how and for what purpose do you apply the total variance explained technique, and what are the important values of this technique for decision making? And so on with each technique.
Authors’ Response:
We have defined all the details and purpose of statistical techniques in Material and Methods section on Page 11, sub-section 3.4 as suggested by the Reviewer in the suggested lines, and highlighted with RED color.
Comments to the Author
Be sure to write the Results section in the past tense.
Author Response:
We have converted all the sentences of the results in the past tense as recommended by the Reviewer, and highlighted with RED color.

Reviewer 3 Report
The issue raised by the authors has been extensively studied in ​​occupational health and other fields of study included in the clinic. However, from the methodological point of view, a new form of analysis is proposed.
The study was cross-sectional and quantitative, and we used a deductive approach to carry out this research. The study was conducted on 873 participants. Respondents were university professors, including permanent, visiting, and adjunct professors. They were selected through an intentional capture of Pakistan's government and private sector universities.
The research work suggests that a linear relationship between spirituality and the work environment is sought. However, the main limitation was not being able to establish a cause-effect relationship. This could be explained by the fact that, from a biostatistical point of view, a data reduction method was used. Although an important connection is demonstrated, it is not possible to know its directionality. Therefore, the term linear relationship could not be possible.
An important problem with the dependent variable is that it has two facets, one being spirituality and the other religiosity. In this sense, although the authors are clear about the differences between the two concepts, the participants may not. Could this cause bias? How was this aspect controlled?
Another observation is the fact that the sample is made up of university academics, the majority with a doctoral or master's degree. I am not sure that the results obtained are similar for workers in other fields in which the organizational environment and stress are different.
Author Response
Reviewer: 3
The issue raised by the authors has been extensively studied in ​​occupational health and other fields of study included in the clinic. However, from the methodological point of view, a new form of analysis is proposed. The study was cross-sectional and quantitative, and we used a deductive approach to carry out this research. The study was conducted on 873 participants. Respondents were university professors, including permanent, visiting, and adjunct professors. They were selected through an intentional capture of Pakistan's government and private sector universities.
Comments to the Author
1) The research work suggests that a linear relationship between spirituality and the work environment is sought. However, the main limitation was not being able to establish a cause-effect relationship. This could be explained by the fact that, from a biostatistical point of view, a data reduction method was used. Although an important connection is demonstrated, it is not possible to know its directionality. Therefore, the term linear relationship could not be possible.
Authors’ Response:
We have corrected the wording in the abstract, discussions, conclusion, and implications, the reviewer’s suggestion is very right & valid it is not the linear relationship. We have corrected the statement and wording, and highlighted with RED color.
Comments to the Author
2) An important problem with the dependent variable is that it has two facets, one being spirituality and the other religiosity. In this sense, although the authors are clear about the differences between the two concepts, the participants may not. Could this cause bias? How was this aspect controlled?
Authors’ Response:
We have comprehensively discussed and differentiated the spirituality, spirituality at work, and religiosity for the general readers on Page No. 6, section 2.1, Page No. 6, 7 & 8, section 2.2 as recommended by the Reviewer, and highlighted with RED color.
Comments to the Author
3) Another observation is the fact that the sample is made up of university academics, the majority with a doctoral or master's degree. I am not sure that the results obtained are similar for workers in other fields in which the organizational environment and stress are different.
Authors’ Response:
We have addressed this issue in limitations & suggested areas the future studies (section 6.2), and we have recommended to the future researchers to replicate the similar model in other field as well. However, this study specifically addressed the problems of universities faculty members. We have also highlighted the material with RED color on Page No. 20.
